# Ferroelectric Tuning of ZnO Ultraviolet Photodetectors

**DOI:** 10.3390/nano12193358

**Published:** 2022-09-27

**Authors:** Haowei Xie, Chenxu Kang, Muhammad Ahsan Iqbal, Xiaoliang Weng, Kewen Wu, Wei Tang, Lu Qi, Yu-Jia Zeng

**Affiliations:** Key Laboratory of Optoelectronic Devices and Systems of Ministry of Education and Guangdong Province, College of Physics and Optoelectronic Engineering, Shenzhen University, Shenzhen 518060, China

**Keywords:** Fe-FET, ZnO, photodetector, CuInP_2_S_6_, P(VDF-TrFE)

## Abstract

The ferroelectric field effect transistor (Fe-FET) is considered to be one of the most important low-power and high-performance devices. It is promising to combine a ferroelectric field effect with a photodetector to improve the photodetection performance. This study proposes a strategy for ZnO ultraviolet (UV) photodetectors regulated by a ferroelectric gate. The ZnO nanowire (NW) UV photodetector was tuned by a 2D CuInP_2_S_6_ (CIPS) ferroelectric gate, which decreased the dark current and enhanced the responsivity and detectivity to 2.40 × 10^4^ A/W and 7.17 × 10^11^ Jones, respectively. This strategy was also applied to a ZnO film UV photodetector that was tuned by a P(VDF-TrFE) ferroelectric gate. Lower power consumption and higher performance can be enabled by ferroelectric tuning of ZnO ultraviolet photodetectors, providing new inspiration for the fabrication of high-performance photodetectors.

## 1. Introduction

Ultraviolet (UV) rays refer to electromagnetic waves with wavelengths in the range of 20–400 nm. UV rays significantly impact human health and the ecological environment, and UV photodetectors are widely used in infrastructure, military facilities, and scientific research. Therefore, it is very important to develop UV photodetectors with high photodetection performance and low power consumption [1,2].

In recent years, many wide-bandgap semiconductor materials have been studied as candidates for next-generation UV photodetectors, among which ZnO [3,4] and GaN [5] are particularly prominent. Compared with other wide-bandgap semiconductors, ZnO has the advantages of a suitable band gap, easy preparation, and low cost [6]. The ZnO-based photodetectors have become a research hotspot [7,8,9,10]. To improve the performance of photodetectors, a localized field was provided to suppress background noise, enhance light absorption, improve electron-hole separation efficiency, amplify photoelectric gain, and expand the detection range of the devices [11,12]. However, applying a sustained gate voltage results in additional power dissipation and increase in device size [13,14]. Therefore, a fabrication strategy for photodetectors with ferroelectric materials as gates was proposed to address the power consumption and size issues [15,16,17]. In the above strategy, a localized field generated by the remanent polarization effect of the ferroelectric material is used to replace the conventional gate [18,19]. Many ferroelectric materials have been utilized to control the carrier concentration and suppress dark current in photodetectors [20,21,22,23]. However, top ferroelectric gated photodetectors cause part of the incident light to be absorbed by ferroelectric material [24]. Therefore, the photodetector device with a back ferroelectric gate needs to be studied [25,26,27].

In this work, we propose a strategy for ZnO nanowire (NW) and thin film UV photodetectors regulated by a ferroelectric back gate. CuInP_2_S_6_ (CIPS) has recently appeared as a promising 2D ferroelectric material [25]. First, we constructed a ZnO NW/h-BN/CIPS/graphene ferroelectric UV photodetector using 2D CIPS as the ferroelectric gate. The electrostatic field generated by the ferroelectric remanent polarization effect of CIPS can effectively suppress the dark current. In addition, the responsivity and detectivity of the device are enhanced to 2.40 × 10^4^ A/W and 7.17 × 10^11^ Jones, respectively. The response time is also reduced. This strategy has also been applied to thin-film devices, where P(VDF-TrFE) is spin-coated as a ferroelectric gate to construct a ZnO/P(VDF-TrFE)/ITO film UV photodetector. Likewise, the presence of the ferroelectric gate improves the photodetection performance of the ZnO film device. This work combines 2D ferroelectrics, NWs, and thin films to construct high-performance ferroelectric photodetectors.

## 2. Experiments

### 2.1. Materials

All materials were purchased commercially without further purification. Graphene, h-BN, and CIPS were purchased from Hefei Kejing Materials Technology Co., Ltd. (Hefei, China). PDMS and Si/SiO_2_ were obtained from PrMat. (Shanghai, China). ZnO target (99.99%) was purchased from Hefei Kejing Materials Technology Co., Ltd. (Hefei, China). P(VDF-TrFE) (70%/30%) was from Beijing HWRK Chem Co., Ltd. (Beijing, China). ITO glass was obtained from Yiyang South China Xiangcheng Technology Co., Ltd. (Hunan, China).

### 2.2. Preparation of ZnO NW Device

First, the marked Si/SiO_2_ substrate was washed with deionized water, acetone, isopropanol, and ethanol for 15 min. Then, the dried Si/SiO_2_ substrate was cleaned by O_2_ plasma. The few-layer graphene on PDMS was exfoliated from the bulk material by mechanical exfoliation and transferred to the Si/SiO_2_ substrate. The exfoliated CIPS was then transferred to the graphene on the substrate. The larger h-BN flakes were chosen to be stacked on top of the CIPS so that the CIPS could be fully covered. Finally, the selected ZnO NWs on PDMS was transferred to the position where the CIPS and graphene overlapped with an optical microscope to form the ZnO NWs/h-BN/CIPS/graphene ferroelectric field effect transistor (Fe-FET) device. ZnO NWs were obtained on the basis of our previous studies [28]. The source and drain electrodes (Ti/Au 5/45 nm) were deposited on both ends of the ZnO NWs, and the gate electrode was deposited on graphene by using electron beam lithography (Raith Pioneer Two), magnetron sputtering, and lift-off techniques.

### 2.3. Preparation of ZnO Film Device

To improve the wettability of the ITO glass, the ITO glass was sequentially cleaned with acetone, isopropanol, and ethanol. Then, the substrate was further cleaned by O_2_ plasma. The prepared P(VDF-TrFE) solution was first filtered through a 0.45 μm nylon-66 syringe filter to remove larger particles in the solution. Afterward, the filtered P(VDF-TrFE) solution was spin-coated (4000 r/min, 60 s) on the cleaned ITO glass under an N_2_ atmosphere. The spin-coated P(VDF-TrFE) film was kept at 80 °C for 15 min and then at 135 °C for 2 h for annealing treatment to improve the crystallinity of the film. Subsequently, ZnO films were deposited on top of thin films by magnetron sputtering under a power of 60 W and working air pressure of 0.5 Pa. Finally, a T-shaped source and drain electrodes (Ti/Au, 5/45 nm) were deposited over the ZnO film by magnetron sputtering. The channel width of the device was 100 μm [10,29].

### 2.4. Characterizations

The X-ray diffraction (XRD) pattern was obtained by the Cu Kα radiation source (λ = 0.15406 nm) (X’Pert Por, Philips, Amsterdam, Netherlands). The λ = 514 nm excitation laser was used to acquire Raman spectra (Senterra, Bruker, Karlsruhe, Germany). The surface topography of the devices was imaged by optical microscopy (RX50M, Soptop, Yuyao, China) and by a scanning electron microscope (SEM Sigma 300, Zeiss, Oberkochen, Germany). The thickness of CIPS and the topological images of the ZnO films were measured by atomic force microscopy (AFM Multimode V, Veeco, NY, USA). The light source for illumination was light-emitting diodes (LEDs) with monochromatic lights of 365 nm, and the light source was manually switched on and off with an interval of 15 s for the response time measurement. The optoelectronic properties of the devices were characterized by the semiconductor analyzer (B2902A, Keysight, Santa Rosa, CA, USA). It is worth noting that the data should have systematic errors less than 5%.

## 3. Results and Discussion

The 3D structure diagram of the ZnO NW Fe-FET device is shown in Figure 1a, in which CIPS and h-BN were used as the ferroelectric layer and insulating layer, respectively. Graphene was used to connect the gate electrode and the CIPS flake so that it was more convenient to deposit the gate electrode after the CIPS flake was completely covered by the h-BN flake. This structure of the prepared device can be observed in the optical microscope image of Figure 1b, in which the width of the electrodes was approximately 5 μm. At the same time, the ZnO NWs were examined by SEM, and the diameter was approximately 300 nm, as shown in Figure 1c. For further determination of the phase of the ZnO NWs, the XRD pattern of the ZnO NWs is shown in Figure 1d. Two main diffraction peaks were observed at 2θ = 31.7° and 36.2°, corresponding to (100) and (101) lattice planes, respectively (JCPDS NO.36-1451). Other peaks 2θ = 34.3°, 47.5°, 62.8°, 66.3°, 67.9°, 69.0°, 72.5°, and 76.9° corresponded to (002), (102), (110), (103), (200), (112), (201), (004), and (202) lattice planes, respectively. No other phase was observed. In the Raman spectrum, the multiple characteristic peaks at 140–290 cm^−1^ of CIPS were due to the S-P-P and S-P-S modes, as shown in Figure 1e. The peaks around 380 cm^−1^ and 410–460 cm^−1^ corresponded to P-P stretching and P-S oscillations, respectively. The peaks generated in the 90–140 cm^−1^ and 300–320 cm^−1^ ranges were caused by cation (Cu^1+^, In^3+^) and anion (P_2_S_6_^4−^) vibrations, respectively [30]. These observed Raman characteristic peaks of CIPS are consistent with previous reports, indicating that CIPS has good crystal quality [31]. The thickness of the CIPS flakes was measured by AFM, and its thickness along the arrow direction was approximately 18.5 nm, as shown in Figure 1f.

In our experiment, we first measured the I_DS_-V_G_ transfer characteristic curve of the ZnO NW Fe-FET device when the source-drain voltage (V_DS_) was 0.1 V, where I_DS_ is the source-drain current, and V_G_ is the gate voltage. A counterclockwise hysteresis loop was obtained as the gate voltage swept from −10 V to +10 V and then from +10 V to −10 V, as shown in Figure 2a. Due to the remanent polarization effect, the device exhibited a storage window of 10.2 V, and the turn-on voltage of the ZnO NW Fe-FET was also shifted from 7 V to −4.8 V under the premise that the CIPS polarity can be reversed. To further analyze the regulation of ZnO NWs by the CIPS bottom gate, the I_DS_-V_DS_ output characteristic curves of the device under different gate voltages (−10 V to +10 V) were plotted, as shown in Figure 2b. As can be seen from the output characteristic curve, I_DS_ significantly increased with gate voltage from −10 V to 10 V.

When a gate voltage of −10 V is applied and then removed, the CIPS will be in a downwardly polarized state and form a vertically upward electrostatic field. Therefore, according to Figure 2a, we defined the device in different CIPS polarization states: the unpolarized state at V_G_ = 0 V is the Fresh state, the reverse polarization state at V_G_ = −10 V is the Down state, and the forward polarized state at V_G_ = +10 V is the Up state. When the device is in the Down state, the bottom gate CIPS creates a high vertical electrostatic field in the opposite direction to the Up state. The I_DS_-V_DS_ curves of the device in these three states showed a strong dependence on the polarization, as shown in Figure 2c. As far as the dark current is concerned, the dark current changed significantly in the Up and Down states with V_DS_ from −1 V to 1 V. In particular, the dark current dropped from 4.75 × 10^−8^ A to 5.25 × 10^−9^ A when the device was in the Down state and V_DS_ at 0.1 V. Moreover, the dark current of the device in the Down state was about two orders of magnitude lower than that in the Up state. The above results indicate clearly that the bottom CIPS ferroelectric gate exhibited a large modulation of the dark current of the device. Under the light wavelength of 365 nm with a light intensity of 50 mW/cm^2^, the ΔI–t characteristic curves of the device were plotted, as shown in Figure 2d and Appendix A, indicating that the photoresponse of the device was very stable in the Down state.

We further explored the responsivity, detectivity, and response time of the device depending on the polarization. Responsivity (R) is related to the photogenerated current, which is defined as the ratio of the photodetector output signal to the input optical power. R can be expressed as R = ΔI/(PA), where ΔI = I_illuminated_ − I_dark_; I_illuminated_ and I_dark_ are the photocurrent and dark current, respectively; P is the power density of the incident light; and A is the light-receiving area. We calculated the responsivity of the three states in Figure 2e, and the R of the device was 2.40 × 10^4^ A/W in the Down state at V_DS_ = 0.1 V with P = 50 mW/cm^2^.

Detectivity (D) is a parameter describing the detection capability of a photodetector, which is closely related to dark current and responsivity. The use of a ferroelectric bottom gate in the device was shown to suppress the dark current to a low level in the experiment. The reduction in dark current was attributed to the high vertical electrostatic field generated by the ferroelectric bottom gate depleting the majority of carriers within the ZnO NWs. Dark current was one of the key factors affecting the detectivity, and its significant change will lead to a change in the detectivity. Detectivity can be calculated from dark current and responsivity, which is given by D = RA^1/2^/(2*e*I_dark_)^1/2^, where R is the responsivity of the photodetector; A and I_dark_ represent the effective illuminated area and dark current, respectively; and *e* is the electron charge. Moreover, it can be calculated that D = 7.17 × 10^11^ Jones in the Down state, 1.81 × 10^11^ Jones in the Fresh state, and 0.20 × 10^11^ Jones in the Up state, as shown in Figure 2e. It is evident that the presence of the ferroelectric bottom gate can effectively control the detectivity of the device.

We further investigated the effect of the bottom-gated ferroelectric effects on the response time of the ZnO NW device. As compared to the other two states, the device in the Down state had faster rise and decay response times of 0.11 and 0.45 s, respectively, as shown in Figure 2f. On the other hand, the response time of the device in the Fresh and Up states were 0.42/5.81 s and 0.6/3.78 s, respectively (Appendix A). It can be seen intuitively that the response time of the device was greatly reduced in the Down state under the regulation of the ferroelectric field. The existence of the bottom ferroelectric field can suppress the recombination of electron–hole pairs inside the device, thereby reducing the response time and enhancing the responsivity and detectivity.

To describe the phenomenon of ZnO NWs modulated by the bottom gate, CIPS ferroelectric polarization in three states at V_DS_ = 0.1 V, the cross-sectional structure, and charge distribution of the ZnO NW device are shown in Figure 3. The arrows represent the polarization direction of the ferroelectric material. In the Fresh state, the bottom gate CIPS was not polarized, as shown in Figure 3a, and the energy band diagram of the ZnO NWs is shown in Figure 3d. The hole–electron pairs inside the ZnO NWs were promoted to separate when irradiated by 365 nm UV light, and the minority carrier holes were greatly excited. The directional movement of the carriers inside the device generated a photocurrent under the bias voltage of V_DS_ = 0.1 V.

The CIPS induced an opposite electrostatic field when the device was applied with a bias voltage of V_G_ = −10 V. Figure 3b shows the bottom ferroelectric gate remnant polarization effect enabled the CIPS to obtain a vertically downward polarization state in the Down state after removing the gate voltage. The energy band diagram of the ZnO NWs in the Down state is shown in Figure 3e. The negative charge facing the ZnO NW channel caused the majority of carrier electrons in the channel to be depleted. The energy bands of the ZnO NWs were bent upward, causing the device to be turned off. The result of the reduced carrier concentration was that the dark current was suppressed to a lower level compared to the Fresh state.

When the device was applied with a bias voltage of V_G_ = +10 V, the induced electrostatic field of the bottom CIPS in the Up state switched to the opposite direction compared to the Down state. After removing the gate voltage, the remnant polarization effect kept the CIPS in a vertically Up state, as shown in Figure 3c. Thus, a large number of electrons accumulated in the ZnO NWs channel, forming an electron accumulation region. In the Up state, due to the accumulation of a large number of carriers, the energy band of the ZnO NWs bent downward, leading to the Fermi level of the ZnO NWs surface rising upwards, as shown in Figure 3f. The carrier concentration in the channel increased, which also increased the dark current compared to that in the Fresh state.

Similar to the above-mentioned 2D devices, we also demonstrated the same phenomenon in the ZnO film Fe-FET device. In this experiment, the ZnO film in the upper layer was modulated by the organic ferroelectric material P(VDF-TrFE) at the bottom. The structure of the ZnO film Fe-FET photodetector composed of bilayers is shown in Figure 4a. The device adopted a sandwich structure in which the ITO glass was used as the gate electrode. The surface of the ZnO film was plated with a pair of T-shaped Au electrodes, and the channel width of the electrodes was 100 μm. Appendix A and Figure 4b depict the AFM images of the spin-coated P(VDF-TrFE) film on the ITO substrate and the ZnO film grown on the surface of P(VDF-TrFE) film. It can be seen that both films exhibited flat and continuous surface morphology. The thickness of the ZnO film was approximately 140 nm for a growth of 30 min. The XRD pattern of the ZnO film displayed in Figure 4c shows the (002) lattice plane diffraction peak at 2θ = 34°, indicating that ZnO preferentially grows along the (002) lattice plane direction. We performed Raman spectroscopy for P(VDF-TrFE). In Figure 4d, the dominant peaks at 841 cm^−1^ and 882 cm^−1^ corresponded to the A_1_ and B_2_ modes, respectively [32,33]. After annealing, P(VDF-TrFE) film also had good crystallinity. Figure 4e shows the ferroelectric hysteresis loop of P(VDF-TrFE) under ±20 V voltage, which exhibited an excellent electroviscous effect [34]. The unpolarized state at V_G_ = 0 V was the Fresh state, the upward polarization state at V_G_ = +20 V was the Up state, and the downward polarization state at V_G_ = −20 V was the Down state of the ferroelectric material P(VDF-TrFE).

By applying different gate voltages, P(VDF-TrFE) film at the bottom of the device can maintain different polarization states. We systematically studied the optoelectronic properties of the ZnO film device under different polarization states (Fresh, Up, and Down states). The I_DS_-V_DS_ curve of the device is shown in Figure 4f. The dark currents of the device in three states were measured by applying a bias voltage from −1 V to 1 V to the electrode on top of the device. We found that in the Down state, the dark current of the device was reduced compared to the Fresh and Up states. In addition, the dark current of the device increased accordingly in the Up state, which is consistent with the experimental expectation.

To further explore the photoresponse of the device, we measured the photocurrent and response time of the device under illumination at a wavelength of 365 nm with a light intensity of 50 mW/cm^2^ and a switching period of 15 s. The ∆I–t characteristic curves of the device at 1 V bias are shown in Figure 4g and Appendix A. The photocurrent of the ZnO film Fe-FET device was not affected by the ferroelectric field, showing a stable and reproducible photoresponse. According to the photocurrent of the device, we also calculated the R and D of the device in Figure 4h, which showed that the R and D of the device increased in the Down state. We also measured the response time of the ZnO film device at a power density of 50 mW/cm^2^. According to Figure 4i, the device had faster response times in the Down state, which was 34.4/153.3 ms. As shown in Appendix A, the other two states also exhibited different response times. The variation in the response time was similar to that of the ZnO NW device. This was again due to the existence of the ferroelectric field at the bottom of the device, which allowed tuning into the performance of the device.

The mechanism of the ZnO film device is similar to that of the ZnO NW device. We also defined the polarization state of the ferroelectric material P(VDF-TrFE) at the bottom of the device. When the polarization of the P(VDF-TrFE) was the Up state, a large number of carrier electrons in ZnO film formed a negative charge accumulation region under the attraction of positive charges. The accumulation of electrons will cause the ZnO energy level to bend downwards and the Fermi level of the surface to rise upwards. Thus, the device was in the “ON” state and the dark current of this state was enhanced compared to the Fresh state. Similarly, the P(VDF-TrFE) was in a vertically downwardly polarized state after being reversely polarized by V_G_ = −20 V. This state depleted the negative charge in ZnO, and the concentration of carriers in the channel decreased. This switched the device to the “OFF” state and lowered its dark current. Table 1 compares the main parameters of our devices in three states. We also compared the main parameters of our ZnO NW device with other common UV photodetectors, as shown in Appendix A [35,36,37,38,39].

## 4. Conclusions

In conclusion, we propose a strategy for ZnO UV photodetectors regulated by a ferroelectric back gate. The modulation of ZnO NW photodetectors was realized by using 2D CIPS flake as the ferroelectric gate. In the Down state, the responsivity of the device was as high as 2.40 × 10^4^ A/W, and the detectivity reached 7.17 × 10^11^ Jones. The organic ferroelectric material P(VDF-TrFE) was also used to modulate the ZnO film photodetector. Our results demonstrated that ferroelectric tuning can reduce the dark current and therefore the device power, which provides a new insight for high-performance UV photodetectors.

## Figures and Tables

**Figure 1 nanomaterials-12-03358-f001:**
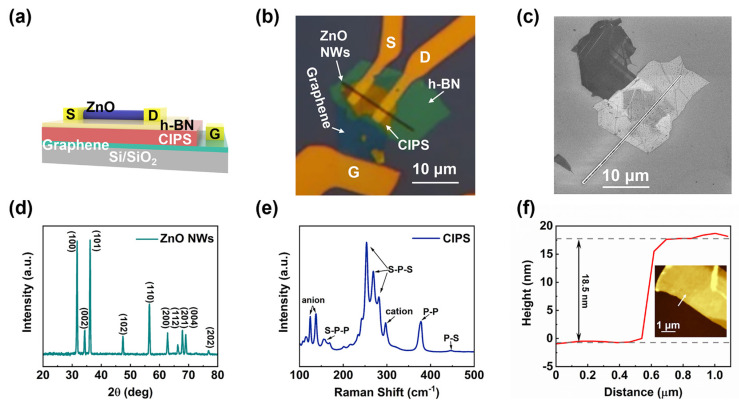
(**a**) Schematic diagram of the ZnO NW Fe–FET device structure, with an insulating layer of h–BN (yellow). (**b**) The optical image of the structure of the ZnO NW Fe–FET device. Graphene, CIPS, h–BN, and ZnO NWs were stacked sequentially; the graphene flakes were connected to the gate (G) electrode and the CIPS flakes; and the source (S) and drain (D) electrodes were connected to both ends of the ZnO NWs. (**c**) SEM image of ZnO NW Fe–FET device before electrode deposition. (**d**) XRD patterns of the ZnO NWs. (**e**) Raman spectra of CIPS flakes. (**f**) Thickness data of CIPS flakes along the white arrow in the AFM image.

**Figure 2 nanomaterials-12-03358-f002:**
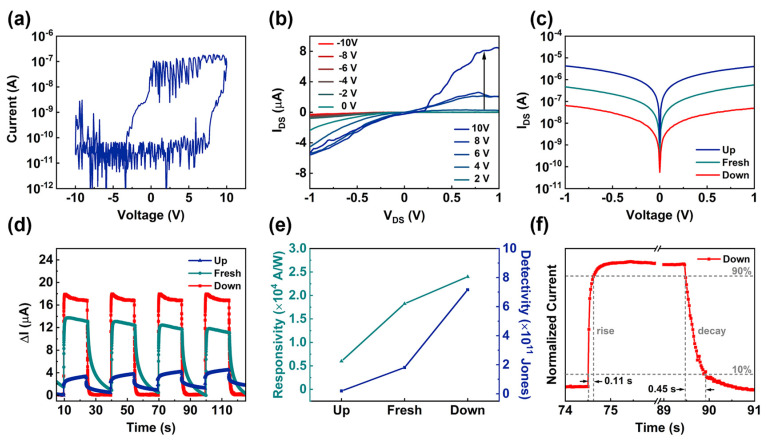
(**a**) Transfer characteristic curves of the ZnO NW Fe–FET device with a CIPS ferroelectric gate at V_DS_ = 0.1 V. (**b**) The I_DS_–V_DS_ output characteristic curves of the device using remanent polarization when CIPS were polarized with −10 V to +10 V gate voltages and the gate voltage was removed. (**c**) The I_DS_–V_DS_ output characteristics with three ferroelectric layer states. (**d**) The ΔI–t characteristic curves with three ferroelectric layer states when V_DS_ = 0.1 V. (**e**) The responsivity and detectivity of the device in three states. (**f**) The response time of the device at a power intensity of 50 mW/cm^2^ in the Down state.

**Figure 3 nanomaterials-12-03358-f003:**
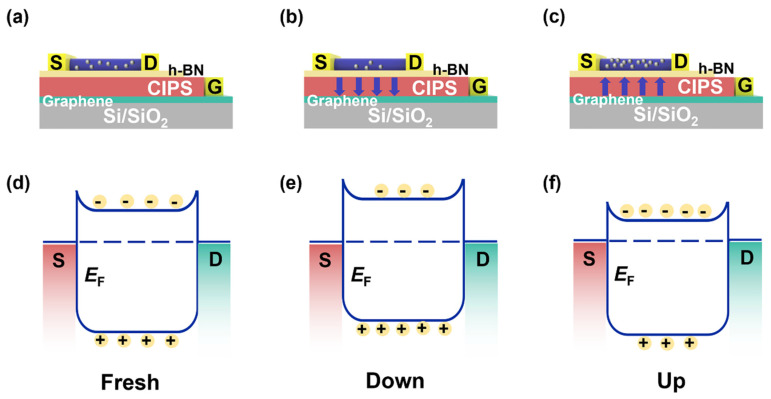
Schematic structure of the ZnO NW Fe–FET device. (**a**) Fresh state, (**b**) Down state, and (**c**) Up state. The corresponding band diagram in the ZnO NW channel when the ferroelectric gate was in (**d**) the Fresh state, (**e**) the Down state, and (**f**) the Up state. *E*_F_ stands for Fermi level.

**Figure 4 nanomaterials-12-03358-f004:**
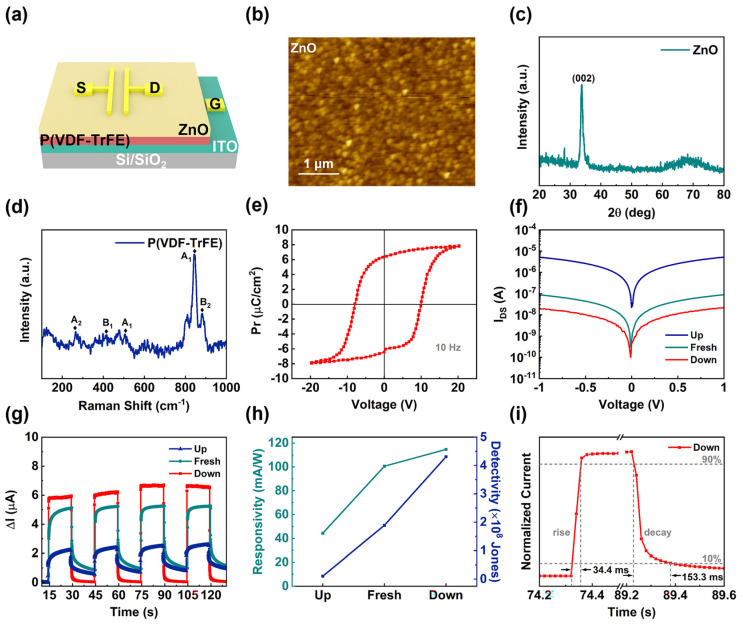
(**a**) Schematic diagram of the structure of the ZnO film Fe–FET device. (**b**) Atomic force microscopy (AFM) topological images of the ZnO films. (**c**) XRD patterns of the ZnO film. (**d**) Raman spectra of the P(VDF–TrFE) film. (**e**) The ferroelectric hysteresis loop of the P(VDF–TrFE) film capacitor, measured with a Sawyer–Tower circuit at a frequency of 10 Hz. (**f**) The I_DS_–V_DS_ output characteristics with three ferroelectric layer states. (**g**) The ∆I–t characteristic curves of the device with three states of a ferroelectric layer at V_DS_ = 1 V. (**h**) The responsivity and detectivity of the device at three ferroelectric layer states. (**i**) The response time of the device at a power intensity of 50 mW/cm^2^ in the Down state.

**Table 1 nanomaterials-12-03358-t001:** Comparison of the main parameters of the devices in three states.

Device	State	Dark Current (A)	Bias Voltage (V)	Responsivity (A/W)	Detectivity (Jones)	Rise/Decay Time (ms)
ZnO NWs	Up	4.19 × 10^−7^	0.1	0.60 × 10^4^	0.20 × 10^11^	600/3780
	Fresh	4.75 × 10^−8^	0.1	1.83 × 10^4^	1.81 × 10^11^	420/5810
	Down	5.25 × 10^−9^	0.1	2.40 × 10^4^	7.17 × 10^11^	110/450
ZnO film	Up	5.24 × 10^−6^	1	0.044	1.08 × 10^7^	4570/10,590
	Fresh	8.86 × 10^−8^	1	0.101	1.89 × 10^8^	715/5180
	Down	2.21 × 10^−8^	1	0.114	4.31 × 10^8^	34.4/153.3

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
