# Peer review of "Ferroelectric Tuning of ZnO Ultraviolet Photodetectors"

_nanomaterials, 2022, doi:10.3390/nano12193358_

Round 1

Reviewer 1 Report

In their work, Xie and collaborators describe their work utilizing ferroelectric polarization to tune the ZnO nanowire and thin film photodetectors, showing a large effect on the detectivity and del(I). I have concerns on this work as detailed below that are not currently addressed sufficiently in the works current form.

1.) The stability of this effect is not addressed in detail. For example, with cycling, the delta(I)-t curves seem to be decaying with time, and it would be very useful to show data for more cycles in order to evaluate how the device performance degrades over time. 

2.) A similar point should be made for Fig. 4g. 

3.) Can the authors comment on how, if at all, changing ferroelectric material to something with a larger remnant polarization would affect measurements? Should this effect saturate at some point? 

4.) The authors should reference a comparison of the improved performance to other common UV photodetector materials. 

5.) The paper needs significant language editing.

Provided these questions are addressed, this work does have potential to make a strong impact within the field of UV photodetector development, but it is important to better contextualize this work within the broader field. 

Reviewer 2 Report

The utility of the proposed devices is not clear.

You claim a huge responsivity for the device based on the ZnO NW. However, the layout is not optimized. The active area is negligible in comparison with the entire device area, so most of the optical power is lost (you cannot focus the input optical power on the NW). You use a huge optical power to obtain a few µA. Moreover, the detectivity seems to be quite low, for such a huge responsivity (and low dark current). Consequently, the results are not credible.

The technology process is very complex and it is not clear the advantage.

Also the lay-out of the device based on the thin-film ZnO seems to be not optimized.

It is also not clear the advantages of using a ferroelectric material instead of a standard gate for the FET.

You say: “ The existence of the bottom ferroelectric 162 field can promote the recombination of electron-hole pairs inside the device, thereby reducing the response time and 163 enhancing the responsivity and detectivity.” Are you sure that you have to increase the recombination to improve  the responsivity? I’d suggest you to read books and papers on photodetectors (in general and photoconductors and ZnO -based devices in particular) and  field effect phototransistors and to try to  better understand the operation of your device.  Do not forget the role of oxygen vacancies.

It is better to read papers and book written by experts in electronics/optoelectronics, not papers written by experts in materials science.

You have not indicated the light source and the set-up for response-time measurement.

I’d suggest to better understand the operation of your device, to re-design the layout and the process in order to improve the light coupling and then publish a paper (only if the device is really useful).

Round 2

Reviewer 1 Report

The authors have addressed my concerns. 

Author Response

We have carefully examined and revised the manuscript.
